# Antagonistic effects of intraspecific cooperation and interspecific competition on thermal performance

Hsiang-Yu Tsai[1,2], Dustin R Rubenstein[3,4], Bo-Fei Chen[1], Mark Liu[1], Shih-Fan Chan[1], De-Pei Chen[1,2], Syuan-Jyun Sun[1], Tzu-Neng Yuan[1], Sheng-Feng Shen[1,2]*

[1]Biodiversity Research Center, Academia Sinica, Taipei, Taiwan; [2]Institute of Ecology and Evolutionary Biology, College of Life Science, National Taiwan University, Taipei, Taiwan; [3]Department of Ecology, Evolution and Environmental Biology, Columbia University, New York, United States; [4]Center for Integrative Animal Behavior, Columbia University, New York, United States

**Abstract** Understanding how climate-mediated biotic interactions shape thermal niche width is critical in an era of global change. Yet, most previous work on thermal niches has ignored detailed mechanistic information about the relationship between temperature and organismal performance, which can be described by a thermal performance curve. Here, we develop a model that predicts the width of thermal performance curves will be narrower in the presence of interspecific competitors, causing a species' optimal breeding temperature to diverge from that of its competitor. We test this prediction in the Asian burying beetle *Nicrophorus nepalensis*, confirming that the divergence in actual and optimal breeding temperatures is the result of competition with their primary competitor, blowflies. However, we further show that intraspecific cooperation enables beetles to outcompete blowflies by recovering their optimal breeding temperature. Ultimately, linking abiotic factors and biotic interactions on niche width will be critical for understanding species-specific responses to climate change.

**\*For correspondence:**
shensf@sinica.edu.tw

**Competing interests:** The authors declare that no competing interests exist.

## Introduction

Recent anthropogenic climate warming makes understanding species vulnerability to changing temperatures one of the most pressing issues in modern biology. A cornerstone for understanding the distribution and associated ecological impacts of climate change on organismal fitness is the concept of the ecological niche, which describes a hyperspace with permissive conditions and requisite resources under which an organism, population, or species has positive fitness (*Hutchinson, 1957*; *Chase and Leibold, 2003*; *Colwell and Rangel, 2009*). More than a half century ago, Hutchinson (*Hutchinson, 1957*) distinguished the fundamental niche—characterized by abiotic environmental measures like temperature and precipitation—from the realized niche—characterized by biological interactions like competition and predation. Although numerous studies since then have used the relationship between a species' distribution and the climate in which it occurs to estimate a species' fundamental niche (*Kearney and Porter, 2009*; *Quintero and Wiens, 2013*; *Gaüzère et al., 2015*), a growing number of studies have also employed theoretical, observational, or experimental approaches to evaluate the detailed mechanisms (e.g. physiological limits) (*Huey et al., 2012*), demographic factors, and species interactions that shape a species' fundamental and realized niches (*Monahan, 2009*; *Nilsson-Örtman et al., 2013*; *Estay et al., 2014*). In general, differentiating between how abiotic factors and biotic interactions influence niche width is critical for understanding species-specific responses to climate change. Ultimately, generating a complete understanding of a species' ecological niche not only requires understanding how abiotic and biotic factors interact to

**eLife digest** Insects, reptiles and many other animals are often referred to as being 'cold-blooded' because, unlike mammals and birds, their body temperature fluctuates with the temperature of their surrounding environment. As a result, many cold-blooded animals are very sensitive to changes in local climate.

Environmental factors, such as temperature and precipitation, as well biotic factors, such as two species competing for food or the presence of a predator, may influence how well an animal performs at different temperatures. However, few studies have examined how both environmental and biotic factors affect the range of temperatures in which a cold-blooded animal is able to survive and reproduce.

When Asian burying beetles reproduce, they lay their eggs around buried animal carcasses that can provide food for their offspring. Previous studies have found that individual burying beetles can cooperate with each other to defend themselves against their main competitor, blowflies, which also lay their eggs on animal carcasses. Here, Tsai et al. used mathematical and experimental approaches to study how blowflies affect the range of temperatures in which burying beetles are able to live under different environmental conditions.

The experiments showed that when blowflies were present, the range of temperatures that burying beetles were able to survive and reproduce in was smaller. Furthermore, the optimal temperature for the burying beetles to live in shifted back, away from that of their competitor. Larger groups of burying beetles were able to survive and reproduce in a greater range of temperatures than smaller groups, even when blowflies were present. This suggests that increasing the amount bury beetles cooperate with each other may make them more resilient to changes in temperature.

The Earth is currently experiencing a period of climate change and therefore it is important to understand how different species of animals may respond to to changing temperatures. These findings reinforce the idea that even a small change in temperature may lead to changes in how different species interact with each other, which in turn influences the ecosystem in which they live.

affect organismal performance and fitness, but also identifying the detailed mechanisms that shape differences in the fundamental and realized niches.

The concept of the thermal performance curve (TPC) was first developed to study how body temperature affects ectotherms' physiological responses and behaviors (*Kearney and Porter, 2009*; *Quintero and Wiens, 2013*; *Gaüzère et al., 2015*). Although the TPC concept has received increasing attention in studies of how warming temperatures influence organismal fitness (*Clusella-Trullas et al., 2011*; *Sinclair et al., 2016*), it has been developed largely independently from niche-based studies. Yet, characterizing the TPC is essentially a way to mechanistically quantify a species' thermal niche. Since the TPC describes the detailed relationship between temperature and fitness, the concept may actually be more informative than that of the thermal niche, which is typically defined as the range of temperatures over which organisms occur in nature (i.e. thermal niche width) (*Huey and Stevenson, 1979*; *Hillaert et al., 2015*). In other words, the TPC concept not only describes thermal niche width, it also quantifies an organism's optimal temperature and how fitness varies with changes in temperature. In contrast, most TPC studies focus largely on what amounts to the fundamental TPC and do not explicitly consider biotic interactions (*Dillon et al., 2010*; *Gunderson et al., 2016*; *Swaddle and Ingrassia, 2017*). The few TPC studies that have considered biotic interactions have almost all focused on predator-prey interactions (*Ockendon et al., 2014*; *Gibert et al., 2016*). To the best of our knowledge, no study has differentiated between fundamental and realized TPCs, nor experimentally quantified a species' realized TPC in the context of interspecific competition, despite the fact that ecologists have long acknowledged that interspecific competition is a key driving force shaping a species' realized niche (*Schoener, 1983*; *Eurich et al., 2018*; *Freeman et al., 2019*). Furthermore, although a few studies have shown that intraspecific cooperation can also help social species expand their realized niche width (*Sun et al., 2014*; *Lin et al., 2019*), little is known about how intraspecific cooperation influences the realized TPCs of social organisms.

Burying beetles (Silphidae, Nicrophorus) are ideal for investigating how social interactions influence both fundamental and realized TPCs because the potentially antagonistic effects of interspecific competition and intraspecific cooperation on the realized TPC can be studied simultaneously. Burying beetles rely on vertebrate carcasses for reproduction and often face intense intra- and interspecific competition for using these limiting resources (*Pukowski, 1933*; *Scott, 1998*; *Rozen et al., 2008*). Competition for carcass access is temperature-dependent because the beetle's main competitors, blowflies (family Calliphoridae), are more abundant and active at higher ambient temperatures (*Sun et al., 2014*). Blowfly maggots also grow faster and digest carcasses more quickly at higher temperatures (*Donovan et al., 2006*; *Kotzé et al., 2016*). In addition to interspecific competition, intraspecific cooperative behavior among beetles may also modulate their realized TPC. Previous studies have shown that cooperative carcass preparation and burial enables beetles to outcompete blowflies and expand their thermal niche to warmer environments (*Sun et al., 2014*; *Chan et al., 2019*; *Chen et al., 2020*; *Liu et al., 2020*).

Here, we extend classic ecological niche theory by introducing the concepts of fundamental and realized TPCs. We first construct a theoretical model by using a hypothetical TPC to predict how interspecific competition influences the width and optimal temperature of realized TPCs in order to provide a general understanding of the relationship between fundamental and realized TPCs. We then describe a series of laboratory and field experiments designed to test the predicted relationship between fundamental and realized TPCs in the Asian burying beetle *Nicrophorus nepalensis* (*Figure 1*). We began our empirical work by measuring breeding performance without interspecific competitors in the laboratory and field to determine *N. nepalensis*'s fundamental TPC. We also

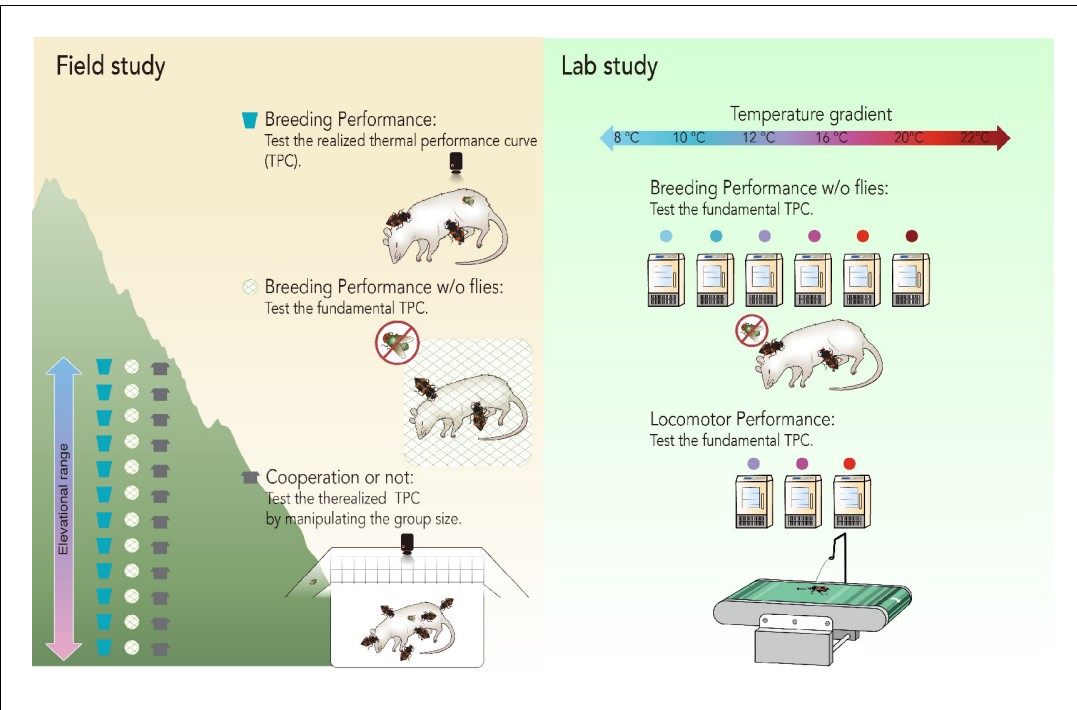

**Figure 1.** A summary of the experiments used to investigate *N. nepalensis*'s fundamental and realized thermal performance curves (TPCs). We examined locomotor and breeding performance in the lab and field without competitors to determine the fundamental TPC. We examined breeding performance in the field in the presence of interspecific competitors to determine the realized TPC. We also manipulated group size in the field to determine how cooperation influenced the realized TPC. The results of these experiments are depicted in *Figures 3–5*.

The online version of this article includes the following figure supplement(s) for figure 1:

**Figure supplement 1.** The relationship between fundamental and realized thermal performance curves between two competing species.

**Figure supplement 2.** A summary of the theoretical and empirical data.

measured the beetle's locomotor ability, which is less likely to be influenced by biotic interactions, at different temperatures to determine the physiological basis of the fundamental breeding TPCs. We then quantified breeding performance in the presence of interspecific competitors, mainly blowflies (*Putman, 1978*; *Putman, 1983*; *Scott, 1994*; *Sun et al., 2014*), to determine the beetle's realized TPC. Finally, we used a group size manipulation in the presence of interspecific competitors in the field to explicitly examine the role of intraspecific cooperation and interspecific competition on the beetle's realized TPC. Experimentally distinguishing between fundamental and realized TPCs will not only serve as a starting point for better understanding the relationships between abiotic and biotic drivers of organismal performance and fitness, but also for better predicting responses to climate change as the earth continues to warm.

## Materials and methods

### Beetle collection and maintenance

Lab experiments were conducted using *N. nepalensis* individuals from a laboratory-reared population. Our stable lab population was established in 2014 from 24 male and 24 female beetles caught near Meifeng, which is 2100 m above sea level on Mt. Hehuan, Taiwan (24°5′ N, 121°10′E). Since establishment, we have supplemented the lab strain with new individuals from the same location every one or two years to avoid inbreeding. We used hanging pitfall traps baited with rotting pork (mean ± SE: 100 ± 10 g) to collect adult beetles in the field. We checked traps and collected beetles on the fourth day after traps were set. In every generation, we established at least 20 families, approximately 600 individuals in total, to maintain the population within the lab. To ensure that beetles in the lab population were unrelated to each other, we always paired beetles collected from different traps. We then put one female and one male in a 20 × 13 × 13 cm box with 10 cm of soil and a rat carcass (75 ± 7.5 g). Approximately two weeks after introducing adult beetles, all of the dispersing larvae that were ready to pupate from each breeding box were collected and allocated to a small, individual pupation box. After roughly 45 days, beetles that emerged from pupae were housed individually in 320 ml transparent plastic cups and fed once a week with superworms (*Zophobas morio*). All breeding experiments were conducted in walk-in growth chambers that imitated natural conditions where the lab population was collected at 2100 m on Mt. Hehuan. Temperature was set to daily cycles between 19°C at noon and 13°C at midnight, and relative humidity was set to 83–100%. We completed all of the laboratory experiments within three generations.

### Breeding performance in the common garden experiment

To investigate breeding TPCs, we conducted solitary pairing experiments in six temperature conditions—8, 10, 12, 16, 20°C and 22°C—in a common garden with no temperature variation in the lab. For each replicate, one male and one female were arbitrarily chosen from different nests to avoid inbreeding. We chose adult beetles that were sexually mature, roughly 2 to 3 weeks after their emergence. Each individual was weighed to the nearest 0.1 mg. We then placed the pair with a mouse carcass (75 ± 7.5 g) under each temperature condition in a transparent plastic container (21 × 13 × 13 cm with 10 cm of soil depth) for two weeks. Cases in which pairs fully buried the carcass and produced offspring were regarded as successful breeding attempts. Cases in which pairs failed to bury a carcass, or they buried it but did not produce offspring, were regarded as failed breeding attempts. The only instance (of 118 replicates) when beetles died during the experiment was excluded from analysis.

### Thermal regulation of locomotor performance

To determine the TPC for locomotor behaviors, we conducted a series of treadmill experiments under three temperature conditions—12, 16°C and 20°C—in a common garden with no temperature variation in the lab. We chose these temperatures because 16°C is the optimal performance temperature for reproduction, and we wanted to further test whether this optimal temperature coincided with optimal physiological function.

We set 72 replicates in total (12°C: 25 replicates; 16°C: 25 replicates; 20°C: 22 replicates). We arbitrarily selected individuals from different nests for replicates at each temperature. The beetles were brought to the experimental chamber one day before data collection began. Monofilament glued to

the pronotum by UV glue attached each beetle to the treadmill, where it was allowed to walk at a stable speed of 150 cm/min. We turned off the treadmill if a beetle's abdomen began to drag or if the beetle started to fly, both behaviors that indicated that the beetle could no longer walk. An individual was tested only once per day. After each experiment, beetles were returned to the transparent container with 3 cm of soil for recovery.

We measured each beetle's pronotum and the ambient temperature during running with a thermal imaging infrared camera (FLIR Systems, Inc, SC305; thermal sensitivity of <0.05°C) at a resolution of 320*240 pixels. Pronotum temperature was measured at the center of the thorax and calculated as the average pronotum body temperature each minute until an individual dragged its abdomen or started flying. The ambient temperature was the average temperature of a $6 \times 6$ cm surface of the treadmill located near where the beetle was tested. The temperature difference was depicted by the difference between the beetle's body and ambient temperatures.

## Breeding performance in the field

Since our previous study showed that blowflies are the beetle's main interspecific competitor (*Sun et al., 2014*), we conducted series of experiments in the field to investigate the breeding TPC with and without interspecific competition. In 2013 to 2016 (May-October), we investigated the natural pattern of *N. nepalensis* reproduction and its breeding success along an elevational gradient from 673 m to 3422 m on Mt. Hehuan in central Taiwan (24°11' N, 121°17' E) that encompasses broadleaf forests at lower elevations and mixed conifer-broadleaf forests at higher elevations. We chose 37 study sites, primarily in natural forests to avoid cultivated or open areas where temperatures are more variable (*De Frenne et al., 2019*) and replicated each treatment at least three times at each site. In each trial, a 75 g (± 7.5 g) rat carcass was placed on the soil to attract beetles and covered with a $21 \times 21 \times 21$ cm (length x width x height) iron cage with $2 \times 2$ cm mesh to prevent vertebrate scavengers from accessing the carcass. We checked each carcass daily until it began to decay due to microbial activity (*Payne, 1965*), was consumed by maggots or other insects, or was buried under the soil by beetles. If burying beetles completely buried the carcass, we checked the experiment after 14 days to determine if third-instar larvae appeared. Cases in which pairs produced third-instar larvae were regarded as successful breeding attempts. Cases in which pairs failed to produce larvae were regarded as failed breeding attempts.

Breeding experiments without blowflies were conducted in the same experimental sites from 2014 to 2017, and 2019 (May-October). The experimental design was the same as that described above, but we used screen mesh above the pots to also keep blowflies out. To record air temperature at every site, we placed iButton devices approximately 120 cm above the ground within a T-shaped PVC pipe to prevent direct exposure to the sun but allow for air to circulate. One male and one female beetle that were reared in the lab were released into the pot to record fundamental breeding performance. After 14 days, we checked the pots to determine whether the burying beetles' third-instar larvae appear. Cases in which pairs fully buried the carcass and produced larvae after 14 days were regarded as successful breeding attempts. Cases in which pairs failed to bury a carcass, or they buried it but did not produce larvae, were regarded as failed breeding attempts. Criteria of data exclusion were the same as the common garden experiments. Three instances (of 178 replicates) when beetles died during the experiment were excluded from analysis.

## The influence of cooperation on thermal performance in the field

To investigate how cooperative behavior influences TPCs, we manipulated the group size of beetles in the field at 38 sites along the elevational gradient. Our experimental device comprised a small plastic container ($21 \times 13 \times 13$ cm with 10 cm of soil) placed inside a large container ($41 \times 31 \times 21.5$ cm with 11 cm of soil). There were several holes on the small container's side wall that allowed beetles to move freely between the two containers. A $2 \times 2$ cm iron mesh was placed around the top of the large container's wall to let flies access the carcass but to keep out larger animals that might scavenge the carcass. Small, non-cooperative groups contained one male and one female, whereas large, cooperative groups contained three males and three females (*Chen et al., 2020*; *Liu et al., 2020*). We captured the local beetles using the same hanging pitfall traps described above, and then conducted two group size treatments at each site. Based on our previous work exploring the natural pattern of arrival times, we released the marked beetles into the experimental

device 1, 2 and 3 days after the trials began at elevations of 1700–2000 m (low), 2000–2400 m (intermediate) and 2400–2800 m (high), respectively (for details, see *Sun et al., 2014*).

Each experiment was recorded by a digital video recorder (DVR) to determine whether *N. nepalensis* successfully buried the carcass. We placed the same temperature measurement device as described above at every site. Cases in which beetles buried the carcass completely and produced larvae after 14 days after were regarded as successful breeding attempts. Cases in which beetles failed to produce larvae were regarded as failed breeding attempts.

## Data analysis

We used generalized linear mixed models (GLMMs) with binomial error structure to compare thermal performance curves among treatments (with/without interspecific competitors; with/without intraspecific cooperation) in the field. The outcome of breeding success (1 = success, 0 = failure) was fitted as a binomial response term to test for differences in the probability of breeding successfully. The variables of interest (i.e. mean daily temperature, type of experimental treatment) were fitted as fixed factors. Other environmental factors (elevation, daily minimum air temperature) were fitted to test the generality of the results. However, since elevation, daily minimum air temperature, and mean air temperature were highly correlated, we only included mean daily temperature in the final model. We also modeled the potential nonlinear effects of the environmental factors by fitting a quadratic regression model and compared the model fit with the linear model. Thus, the thermal performance curves (TPC) were determined statistically by the GLMM. To account for repeated sampling in the same plot, we set the field plot ID as a random factor (coded as 1|plot ID) in the R package lme4 (*Bates et al., 2014*). We also included year as a random factor to account for sampling at different time points (See *Source code 1* for further details).

We used a general linear models (GLMs) to determine *N. nepalensis*'s breeding rate and locomotor performance in the lab. The outcome (1 = success, 0 = failure) was fitted as a binomial response term to test the difference in the probabilities of interest (burial or non-burial) under different temperatures. For locomotor performance, the outcome (1 = flying, 0 = not flying) was fitted as a binomial response term to test for a difference in the probability of interest (flying or not) under different temperatures conditions. The relationship of temperature difference (body temperature minus ambient temperature) was fitted as a Gaussian response to different temperature conditions. (See *Source code 1* for further details).

Finally, the optimal temperature ($T_{optimal}$) of the TPC was calculated by taking the derivative of the regression line that described the relationship between temperature and the likelihood of breeding successfully. In other words, $T_{optimal}$ was estimated from the unimodal statistical model of the TPC. To estimate TPC breadth, we calculated the 95% confidence interval of the regression line. The boundaries of the TPC were the points that there was not significant difference between regression lines and zero. All statistical analyses were performed in the R v3.0.2 statistical software package (*R Development Core Team, 2018*). (See *Source code 1* for further details).

## Theoretical model

We employed the following commonly-used thermal performance curve (TPC) function (*Deutsch et al., 2008*; *Vasseur et al., 2014*) to describe the fundamental TPCs of two competing species with strategies similar to those in our empirical system of beetles and blowflies. A low-temperature thermal specialist species resembles the burying beetle's breeding thermal performance (*Tsai et al., 2020*), whereas a high temperature generalist species that has a similar life history to the blowfly (*Figure 1a*):

$$P(T) = \left\{ \begin{array}{l} \exp(-(T - T_{opt})/2\sigma_p)^2), \ when \ T \leq T_{opt} \\ 1 - ((T - T_{opt})/(T_{opt} - T_{max}))^2, \ when \ T_{opt} \end{array} \right\} \tag{1}$$

where $T$ is environmental temperature, $\sigma_p$ is the shape parameter describing the steepness of the curve at the lower end, $T_{opt}$ is the optimal environmental temperature at which organisms have their highest performance, and $T_{max}$ is the upper critical temperature. We assumed that performance becomes zero when $T > T_{max}$.

Since environmental conditions also directly influence a species' average performance, we used a Gaussian function to describe the chance of encountering a particular temperature:

$$f(T|T_{mean}) = (2\pi V)^{-0.5\exp\left(-(T_s-T_m)^2/2V\right)} \tag{2}$$

where $f(T|T_{mean})$ represents the probability of getting $T$ given $T_{mean}$, $T_{mean}$ represents the mean environmental temperature, and $V$ describes the environmental temperature variability. We combined equations (1) and (2) to obtain the temperature-weighted performance function:

$$w(T) = \int_{-\infty}^{+\infty} [P(T)f(T|T_{mean})]dt \tag{3}$$

by integrating the product of performance and probability of the temperature across all environmental temperatures. We then used the relative performance of the two species (i.e. $w_S(T)/w_G(T)$ and $w_G(T)/w_S(T)$) to represent the realized TPCs of specialist and generalist species, respectively.

We began by addressing how interspecific competition influences the realized TPC of a focal species, finding that when a low temperature specialist species (e.g. burying beetle) competes with a high temperature generalist species (e.g. blowfly), the optimal temperature of the realized TPC of the thermal specialist shifts towards a lower temperature and the width of the TPC decreases (*Figure 2b*). In other words, our model predicts that the optimal temperature of the realized TPC will decrease to below that of the optimal of fundamental TPC when a low temperature specialist competes with a high temperature generalist. To make the theoretical framework complete, we also explored the scenario of a high temperature specialist competing with a low temperature generalist. We found that if a high temperature specialist species competes with the low temperature generalist species (*Figure 1—figure supplement 1a*), the optimal temperature of the realized TPC of the thermal specialist shifts towards a higher temperature and the width of the realized TPC decreases (*Figure 1—figure supplement 1b*), which suggests that a shift in the optimal realized TPC away from the optimal temperature of the competing species is a general result. (See *Source code 2* for further details).

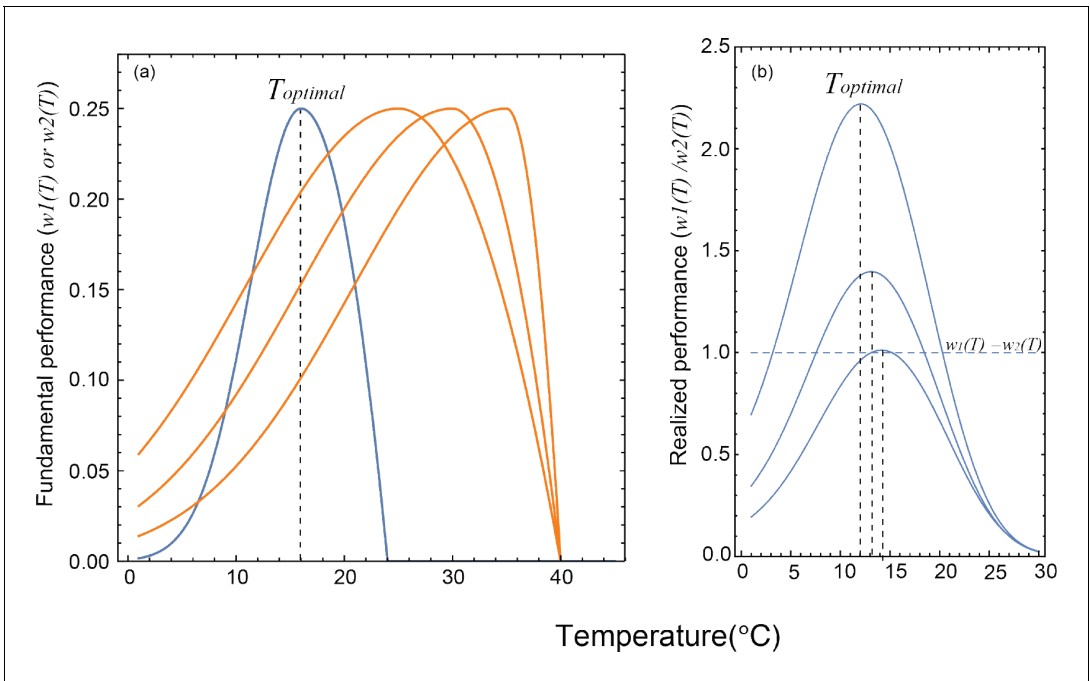

**Figure 2.** The (a) fundamental thermal performance curves (TPCs) used in the theoretical model to generate (b) the predicted realized TPCs. Blue lines represent the TPCs of the low temperature specialist species, whereas orange lines depict the high temperature generalist species. $T_{optimal}$ indicates the optimal temperature of the realized TPCs, as shown by vertical dashed lines. The horizontal dashed line represents the relative performance of the two species, which are equal to each other.

## Results

We first explored *N. nepalensis*'s fundamental breeding TPC in a controlled lab environment. We found that the probability *N. nepalensis* breeding successfully changed unimodally with ambient temperature (*Figure 3a*, GLM, $\chi^2_2$ = 26.29, p < 0.001, n = 117). Accordingly, we found that the beetle's optimal breeding temperature or fundamental TPC—defined as the mean temperature at which breeding success was highest (calculated from our GLM)—was 15.6°C. To determine the physiological basis of this optimal breeding TPC, we measured locomotion ability at different temperatures by performing a treadmill running experiment. We found that beetles had a greater likelihood of flying at 16°C while running at a stable speed on the treadmill (*Figure 3b*, GLM, $\chi^2_2$ = 13.22, p = 0.001,

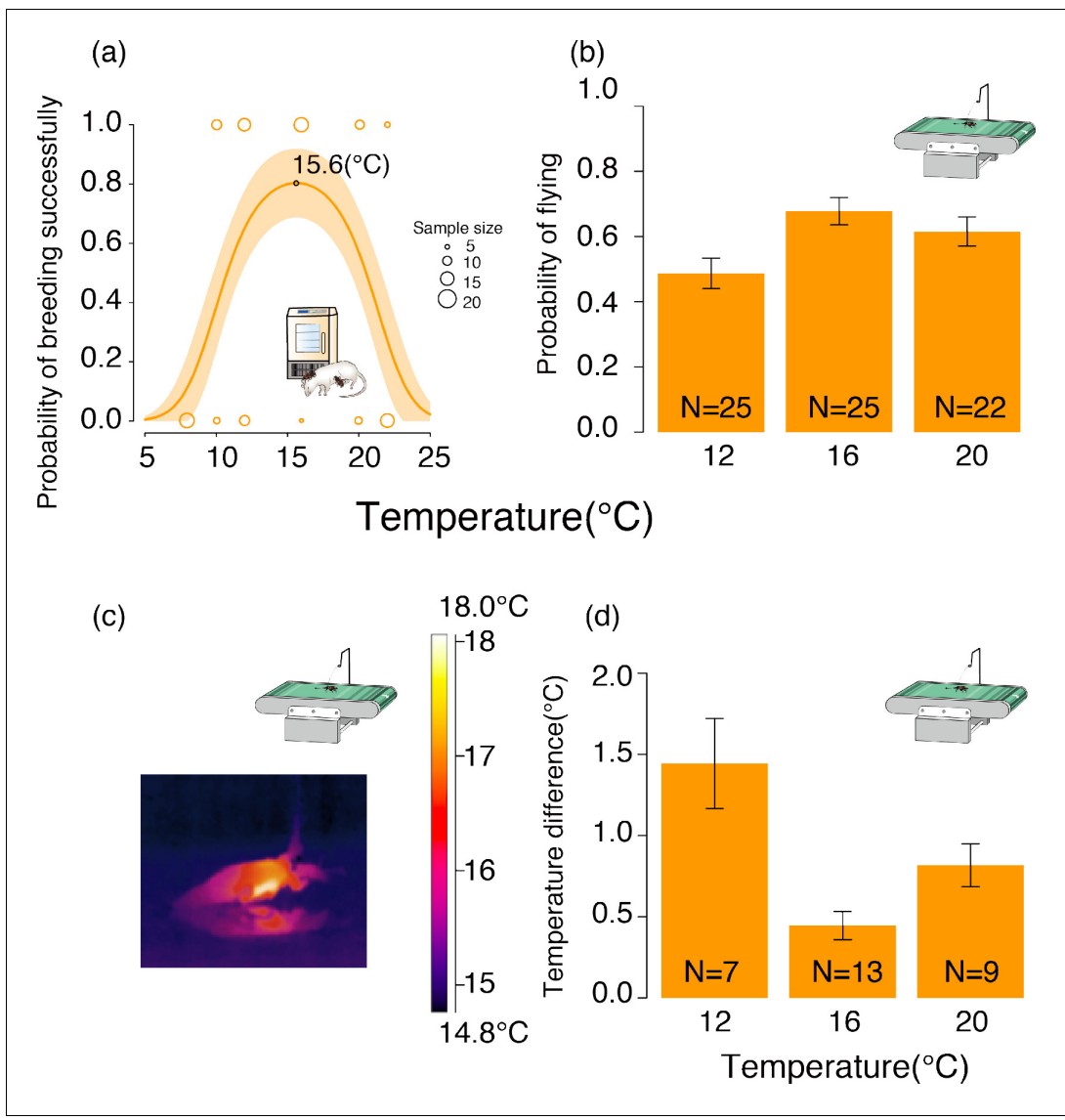

**Figure 3.** The fundamental thermal performance curve (TPC) of *N. nepalensis* represented by the relationship between the probability of breeding successfully and temperature in the lab. (a) The probability of breeding successfully at different temperatures in the lab in the absence of competitors. The shaded orange area depicts the 95% confidence interval. The optimal temperature of the TPC, which is calculated by taking the derivative of the regression line, is indicated ($\chi^2_2$ = 26.29, p < 0.001, n = 117). (b) The probability of flying in the three different ambient temperature conditions ($\chi^2_2$ = 13.22, p = 0.001, n = 72) (c) Infrared thermal image of *N. nepalensis* running on the treadmill, and (d) the temperature difference between the body temperature of beetles that started flying and the ambient temperature in the three temperature conditions while beetles were running on the treadmill at a stable speed of 150 cm/min ($\chi^2_2$ = 23.18, p < 0.001, n = 29).

n = 72). In other words, *N. nepalensis* took less energy to raise its body temperature enough to begin flying at 16°C than at other temperatures (***Figure 3c and d***, GLM, $\chi^2_2$ = 23.18, p < 0.001, n = 29).

Next, we investigated *N. nepalensis*'s realized and fundamental breeding TPCs by studying breeding performance along an elevational gradient (1600 to 2800 m above sea level). As predicted by our model, in the presence of interspecific competitors (blowflies) in the wild, the optimal breeding temperature (i.e. the realized TPC) of *N. nepalensis* was roughly 13.1°C, which is lower than the optimal temperature in the lab in the absence of blowflies (i.e. the fundamental TPC) (***Figure 4a***, GLMM, $\chi^2_2$ = 16.56, p < 0.001, n = 343). Intriguingly, when excluding blowflies and removing the threat of intraspecific competition in the field, the optimal breeding temperature of *N. nepalensis* increased to approximately 14.6°C, such that the realized TPC began to approach the fundamental TPC in the absence of blowflies, ultimately becoming broader than when in the presence of interspecific competitors (***Figure 4b***, GLMM, $\chi^2_2$ = 16.08, p < 0.001, n = 175). Thus, our experiment confirmed the causal relationship between interspecific competition and the shift in the realized TPC under natural conditions.

Since our previous study found that *N. nepalensis* will cooperate at carcasses to compete against blowflies, particularly in warmer environments (***Sun et al., 2014***), we predicted that a group of *N. nepalensis* in a warm environment would have a better chance of expanding its realized TPC towards the fundamental TPC than would an individual pair. To test this prediction, we performed a group size manipulation to determine the realized TPCs of cooperative groups and solitary beetles. We found that *N. nepalensis* in cooperative groups had an optimal breeding temperature (i.e. realized TPC) of 15.6°C, which is identical to their optimal temperature from the fundamental TPC in the lab. In contrast, the optimal breeding temperature of solitary pairs was 14.1°C, similar to that of the realized TPC of 14.6°C in the field (***Figure 5***, group size × temperature interaction, $\chi^2_2$ = 6.84, p = 0.033, n = 328; for large groups, $\chi^2_2$ = 9.40, p = 0.009, n = 162; for small groups, $\chi^2_2$ = 18.42, p < 0.001, n = 166). These results suggest that beetles that cooperate are able to expand their realized TPCs such that they converge on their fundamental TPCs, whereas those do not cooperate have divergent realized and fundamental TPCs.

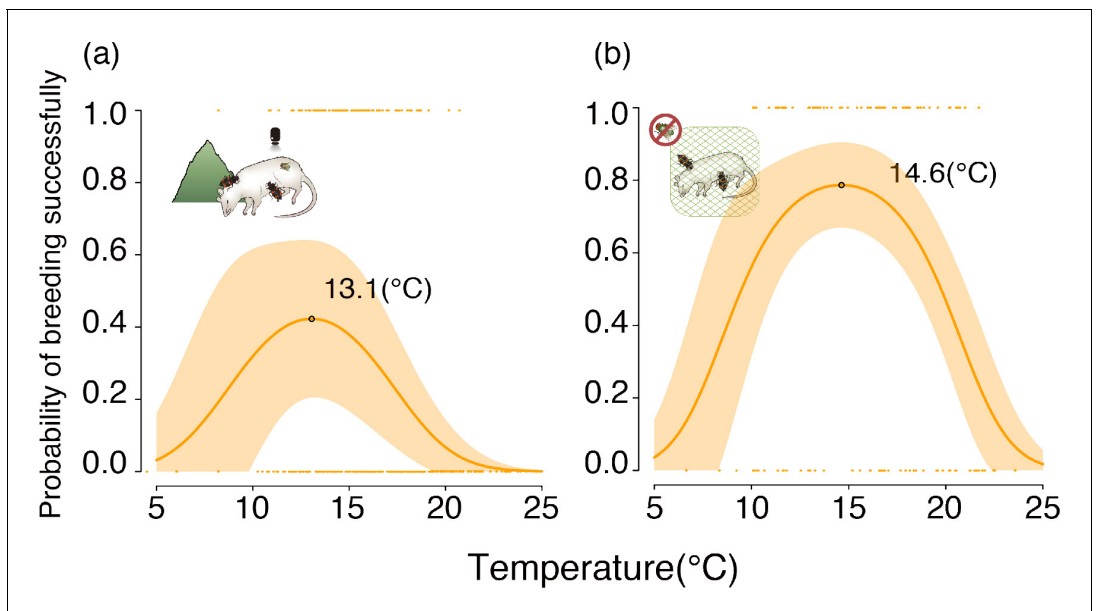

**Figure 4.** Realized thermal performance curves of *N. nepalensis* represented by the relationship between the probability of breeding successfully and temperature in the field. The probability of breeding successfully along the temperature gradient (**a**) with the potential for interspecific competition ($\chi^2_2$ = 16.56, p < 0.001, n = 343) and (**b**) in the absence of interspecific competition. The shaded orange area depicts the 95% confidence interval ($\chi^2_2$ = 16.08, p < 0.001, n = 175). The optimal temperatures of the TPCs, which were calculated by taking the derivatives of the regression lines, are indicated. Pictures correspond to experiments detailed in ***Figure 1***.

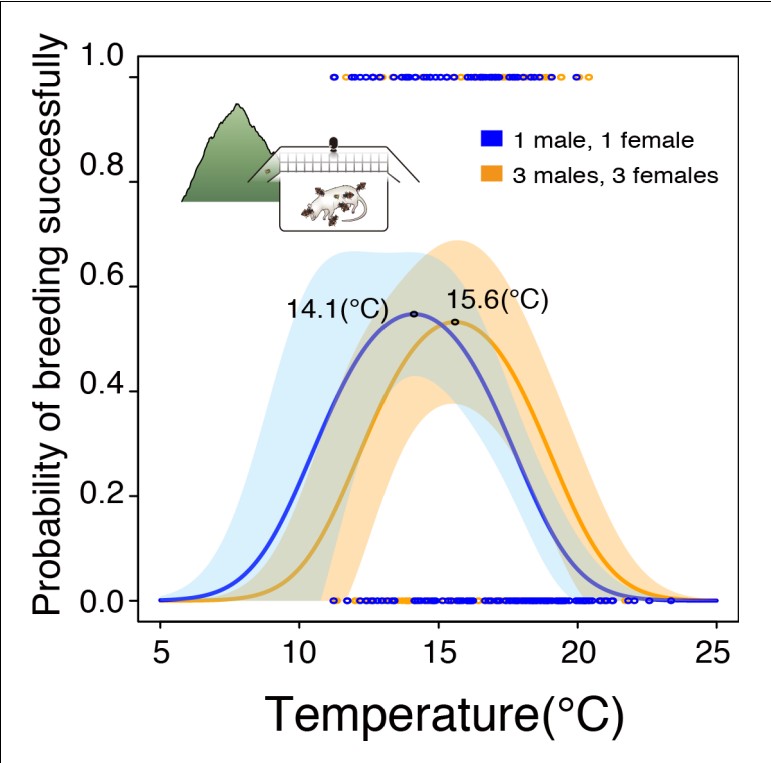

**Figure 5.** Realized thermal performance curves of *N. nepalensis* represented by the relationship between the probability of breeding successfully and temperature in the group size treatment (orange circles, orange line) and the solitary control (blue circles, blue line). The shaded areas depict 95% confidence intervals (group size ×temperature interaction, $\chi^2_2$ = 6.84, p = 0.033, n = 328; for large groups $\chi^2_2$ = 9.40, p = 0.009, n = 162; for small groups, $\chi^2_2$ = 18.42, p < 0.001, n = 166). The optimal temperatures of the TPCs, which were calculated by taking the derivatives of the regression lines, are indicated. Pictures correspond to experiments detailed in *Figure 1*.

## Discussion

By combining the concepts of fundamental and realized niches from ecological niche theory with the that of the thermal performance curve (TPC), we found that a species' realized thermal performance curve is likely to change in time and space in response to biotic factors such as interspecific competition (see *Figure 1—figure supplement 2* for the summary). Our theoretical model suggests that if thermal specialist species compete with thermal generalist species adapted to higher or lower temperatures, the optimal performance temperature of specialists will decrease or increase, respectively. Our empirical results examining competition between burying beetles (thermal specialists) and blowflies (thermal generalists) for access to carcasses support this theoretical prediction, finding that blowflies force the beetle's optimal breeding temperature lower and the realized TPC narrower. Intriguingly, our experiment also showed that intraspecific cooperation in this facultatively social species not only enables beetles to overcome interspecific competition (*Sun et al., 2014*; *Shen et al., 2017*; *Lin et al., 2019*), but to better match their fundamental and realized TPCs (from 14.1˚C to 15.6˚C). Therefore, the mechanism that enables cooperative beetles to expand their range to lower elevations relative to non-cooperative beetles (*Sun et al., 2014*; *Liu et al., 2020*) appears to be their ability to align their fundamental and realized thermal niches.

The idea that interspecific competition will reduce the realized niche width of a species is well-accepted in ecology. However, our study further suggests that a more mechanistic understanding of how interspecific competitors affect the optimal temperature performance of species will be critical for understanding how climate change affects species' vulnerability. Since it is generally assumed in studies of macroecology and climate change that thermal performance is largely influenced by physiology, a single function is often used to describe a species' thermal performance curve

(*Sinclair et al., 2016*). However, if biotic interactions are key to indirectly influencing the thermal performance of a species, as we have shown here, the realized TPC of a species is likely to change in time and space and should not be described by a single function to represent the thermal performance of a species.

Integrating the idea of TPCs into the ecological niche concept helps bridge two rich, but largely independent, traditions of studying thermal adaptation. By simply recognizing the concept of realized TPCs, it becomes clear that we know little about how realized and fundamental TPCs differ in most species. We show that the realized TPC provides a way to quantify how temperature mediates species interactions, which also influence organismal fitness. Thus, the realized TPC extends the realized thermal niche concept, which only considers the temperature ranges in which a species can occupy or breed successfully (*Huff et al., 2005*; *Rehfeldt et al., 2008*) by considering how abiotic factors (i.e. climate) additionally indirectly affect fitness by driving biotic interactions like species competition. For example, a greater population size can facilitate intraspecific cooperation because there are more individuals to form groups quickly to outcompete interspecific competitors, as we found in burying beetles at lower elevations. We predict that Allee effects—when higher population density has a positive effect on individual fitness and population growth rate until it reaches the maximum (*Allee, 1931*; *Courchamp et al., 1999*; *Stephens and Sutherland, 1999*)—will likely occur in *N. nepalensis* in warmer environments. Therefore, conserving high population densities, especially at lower elevations, will be crucial for *N. nepalensis* to compete against blowflies under increased climate warming. By examining the relationship between cooperative behavior and interspecific competition, our study thus helps understand the pressing issue of how habitat destruction affects the vulnerability of social organisms to climate change (*Travis, 2003*). When population density influences the likelihood of intraspecific cooperation in social species, habitat destruction will not only decrease habitat availability but also weaken a species' competitive ability against interspecific competitors, which in turn will lower the realized thermal performance of social organisms.

Our study has implications beyond interspecific competition in insects. Many classic studies of TPCs investigate how changes in body temperature influence physiological or behavioral performance (*Chen et al., 2003*; *Zhang and Ji, 2004*; *Huey et al., 2012*). Body temperature is often assumed to be the same as the environmental temperature in ectotherms. However, accumulating evidence suggests that many ectotherms can at least partially regulate their own body temperature behaviorally or physiologically (*Heinrich, 1993*; *Tattersall et al., 2016*; *Clarke, 2017*). Thus, to identify fundamental TPCs, it is crucial to perform both lab and field experiments in order to understand the exact relationship between body and ambient temperature, as well as how realistic environmental conditions (e.g. temperature variation, humidity, or precipitation) influence the fundamental TPC. For example, TPCs are often considered to be left-skewed for physiological reasons (e.g. the response of thermoregulation proceeds faster at higher temperatures) (*Woodin et al., 2013*; *Huey and Pianka, 2018*), but our results and those of many other studies have shown that TPCs can be diverse in their shape (*Dell et al., 2011*; *Monaco et al., 2017*). Although we show that variation in the shape of TPCs can be due to interspecific competition, other types of biotic interactions, including mutualistic and host-parasites interactions (*Cohen et al., 2017*), should also be carefully considered and compared when quantifying fundamental TPCs. Ultimately, separating and experimentally quantifying fundamental and realized thermal performance in the field and lab will be critical for understanding how both biotic and abiotic factors interact to influence organismal fitness, particularly in an era of rapid climate change.

The concept of the TPC has received renewed interest because the earth has been warming rapidly for the past few decades. Yet, apparent gaps exist between studies of physiological function and those examining fitness consequences in changing environments. Our study shows that employing the concepts of fundamental and realized TPCs can help us predict the ecological impacts of climate change, especially because environmental change will likely reshuffle ecological communities and alter the strength of species interaction (*Alexander et al., 2016*). The importance of biotic interactions in shaping species distributions and community composition is intuitively obvious, yet historically has been difficult to quantify. We believe that the concept of realized TPCs can help fill this important knowledge gap and, ultimately, deepen our understanding of the ecological impact of climate change.

## Acknowledgements

This work is funded by Ministry of Science and Technology, Taiwan (103–2621-B-001–003 -MY3 and 101–2313-B-001–008 -MY3 to S-F S) and Academia Sinica (AS-SS-106–05 and AS-IA-106-L01 to S-FS). DRR was supported by the US National Science Foundation (IOS-1656098).

## Additional information

### Funding

| Funder | Grant reference number | Author |
|---|---|---|
| Ministry of Science and Technology, Taiwan | 103-2621-B-001 -003 -MY3 | Sheng-Feng Shen |
| National Science Foundation | IOS-1656098 | Dustin Reid Rubenstein |
| Ministry of Science and Technology, Taiwan | 101-2313-B-001 -008 -MY3 | Sheng-Feng Shen |
| Academia Sinica | AS-SS-106-05 | Sheng-Feng Shen |
| Academia Sinica | AS-IA-106-L01 | Sheng-Feng Shen |

The funders had no role in study design, data collection and interpretation, or the decision to submit the work for publication.

### Author contributions

Hsiang-Yu Tsai, Conceptualization, Data curation, Formal analysis, Validation, Investigation, Visualization, Methodology; Dustin R Rubenstein, Conceptualization, Funding acquisition, Investigation, Writing - review and editing; Bo-Fei Chen, Mark Liu, Shih-Fan Chan, De-Pei Chen, Syuan-Jyun Sun, Tzu-Neng Yuan, Investigation, Methodology; Sheng-Feng Shen, Conceptualization, Resources, Supervision, Funding acquisition, Validation, Investigation, Writing - original draft, Project administration

### Author ORCIDs

Dustin R Rubenstein ⓘ https://orcid.org/0000-0002-4999-3723
Bo-Fei Chen ⓘ http://orcid.org/0000-0003-3005-8724
Sheng-Feng Shen ⓘ https://orcid.org/0000-0002-0631-6343

### Decision letter and Author response

Decision letter https://doi.org/10.7554/eLife.57022.sa1
Author response https://doi.org/10.7554/eLife.57022.sa2

## Additional files

### Supplementary files

- Source code 1. Statistical source code.
- Source code 2. Mathematica code of the theoretical model.
- Transparent reporting form

### Data availability

All data analysed during the study are available in Dryad.

The following dataset was generated:

| Author(s) | Year | Dataset title | Dataset URL | Database and Identifier |
|---|---|---|---|---|
| Shen SF, Tsai HY, Rubenstein DR, Chen BF, Liu M, Chan SF, Chen DP, | 2020 | Source data for: Antagonistic effects of intraspecific cooperation and interspecific competition on thermal performance | https://datadryad.org/stash/dataset/doi:10.5061/dryad.w0vt4b8nw | Dryad Digital Repository, 10.5061/dryad.w0vt4b8nw |

Sun SJ, Yuan TN

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
