## [Decision Letter]

**Acceptance summary:**

This paper combines theory with lab and field experiments to address an interesting and important question: how does competition between species, and its interaction with cooperation within species, influence the thermal performance curve of animals (burying beetles) and thus their potential distribution? The resulting work is novel and provides important insights into the role that biotic interactions play in shaping the ecological niche of a population and thereby how populations will respond to climate change.

**Decision letter after peer review:**

Thank you for submitting your article "Antagonistic Effects of Intraspecific Cooperation and Interspecific Competition on Thermal Performance" for consideration by *eLife*. Your article has been reviewed by three peer reviewers, including Samuel L Díaz-Muñoz as the Reviewing Editor and Reviewer #1, and the evaluation has been overseen by Christian Rutz as the Senior Editor. The following individual involved in the review of your submission has agreed to reveal their identity: Per Smiseth (Reviewer #2).

The reviewers have discussed their reviews with one another, and the Reviewing Editor has drafted this decision to help you prepare a revised submission.

Summary:

This manuscript describes impressive work that addresses an interesting and important question: how does interspecific competition, and its interaction with intraspecific cooperation, influence the thermal performance curve of burying beetles? The authors answer this question using a combination of lab and field experiments, together with theory, to show that temperature-dependent breeding performance in burying beetles varies with the density of competing blowflies and the number of cooperative beetles. This paper combines two currents in the field: (1) the use of physiology to understand niches and how species distributions may track climate change; and (2) an examination of how inter- and intra-species interactions may change realized niches. The work is novel and likely to provide important new insights into the role that biotic interactions play in shaping the thermal niche of a population, and thereby how populations will respond to climate change. In sum, this manuscript has the potential to become a landmark paper in the field, because it integrates important research currents with formal theory and empirical data.

In preparing your response and manuscript modifications, please pay special attention to the Essential revisions section below. There are a number of them, but the majority pertain to added detail in the manuscript, and should not require new data. There was concern that the manuscript is short on detail on some key aspects of the methodology, with potential implications for the interpretation of the results.

Essential revisions:

1) The authors should indicate more clearly the model they used and the parameters they estimated to obtain TPCs from their data (e.g., Figure 3B). They state that they used a GLMM to model a binomial response to temperature, but the nice humped shape of the curve for the probability of success requires a more complex statistical model than is suggested in the data analysis section. If they are fitting the theoretical model described in Equation 1, then they should give the parameter estimates for Topt, Tmax and σ. If they are fitting Equation 3, they should provide parameter estimates for that equation. And, if they are estimating the parameter Topt statistically, then it is unclear why they would need to derive Topt by taking the derivative of the curve generated with the estimated Topt.

2) Why does the text referring to the TPC in Figure 3A provide Chi-square and p-values? What is the null hypothesis here? Is it that breeding performance does not vary with temperature? If so, that should be stated clearly.

3) The authors mention that they included environmental covariates, elevation and daily minimum temperature, but did not indicate whether these had any influence on breeding success. Also, was there any attempt to account for year in the analysis?

4) The fact that the field group size experiments match the realized TPC in the field and the fundamental TPC in the lab exactly is curious, given how variable experiments (especially those in the field) can be. Perhaps the use of more significant figures and listing the uncertainty around these estimates would allay any concerns from readers.

5) Related to the above point: The code underlying the analyses should be included so readers can evaluate how the results were arrived at, also addressing concerns above regarding the model. (If it is not already planned to be included in the Dryad Repository, especially given the nice data break down by figure).

6) How is competition between blowflies and beetles modeled in the theoretical section? There must be some equation used to represent how access to a carcass varies with temperature due to the activity of blowflies, but we couldn't see the equation or any reference to it.

7) Materials and methods: How many traps were placed within the altitudinal range, and how far apart were they?

8) Origins of the lab strains of beetles used in the laboratory experiments: Please provide more details on how many lab strains were used in the experiments and the extent to which these lab strains originated from separate populations with distinct thermal conditions. Currently, the manuscript simply states that each lab strain derived from 2 males and 2 females caught in the field.

9) Breeding performance in the common garden experiment: Please explain why males and females from different populations were paired up in this experiment, and whether this approach had any implications for the interpretation of the results. If lab strains originated from separate populations with distinct thermal conditions, and there is local adaptation to these thermal conditions, could this approach alter the offspring's TPC compared to that of natural populations? If so, could this have implications for the interpretation of your results, such as for comparisons of breeding performance under laboratory and natural conditions?

10) Thermal regulation of locomotor performance: This experiment was conducted by beetles from different lab strains. Given that each lab strain derived from 2 males and 2 females caught in the field, there is likely to be some inbreeding in each lab strain. Could this have implications for the interpretation of these results, such as for comparisons of TPC for locomotory and breeding performance?

11) Breeding performance in the field: Our understanding is that different treatments were replicated (at least to some degree) in different years. The treatment 'with interspecific competition' was replicated in 2014 and 2015, whilst the treatment 'without interspecific competition' in 2014 and 2019? What implications, if any, could this have for the interpretation of your results?

12) The Introduction (like the Abstract) doesn't emphasize, or rather weave together, the two important currents of niche research (TPC and cooperation, as mentioned above in the summary) that are addressed explicitly and simultaneously in this paper. As in the Abstract, it may be helpful to highlight cooperation (or competition/cooperation balance) earlier in the paper. Something akin to the gem of a paragraph that starts with "Integrating the idea of TPCs into the ecological niche concept helps bridge two rich, but largely independent, traditions of studying thermal adaptation." in the Discussion. A similarly strong argument earlier in the Introduction can help tie together the TPC/behavior currents.

---

## [Author Response]

Essential revisions:1) The authors should indicate more clearly the model they used and the parameters they estimated to obtain TPCs from their data (e.g., Figure 3B). They state that they used a GLMM to model a binomial response to temperature, but the nice humped shape of the curve for the probability of success requires a more complex statistical model than is suggested in the data analysis section. If they are fitting the theoretical model described in Equation 1, then they should give the parameter estimates for Topt, Tmax and σ. If they are fitting Equation 3, they should provide parameter estimates for that equation. And, if they are estimating the parameter Topt statistically, then it is unclear why they would need to derive Topt by taking the derivative of the curve generated with the estimated Topt.

The TPCs were determined statistically by GLMM. We compared the model fits of linear and quadratic GLMMs. Since the quadratic models fit better than the linear ones, we used the quadratic model to represent the TPC of burying beetles. Consequently, taking the derivative is the easiest way to obtain Topt. We have modified the text accordingly to avoid confusion:

“We also modeled the potential nonlinear effects of the environmental factors by fitting a quadratic regression model and compared the model fit with the linear model. […] We also included year as a random factor to account for sampling at different time points (See Source Code 1 for further details).”

2) Why does the text referring to the TPC in Figure 3A provide Chi-square and p-values? What is the null hypothesis here? Is it that breeding performance does not vary with temperature? If so, that should be stated clearly.

Yes, the null hypothesis is breeding performance does not vary with temperature as shown by the Chi-square and p-values. We have modified the sentence as follows:

“We found that the probability *N. nepalensis* breeding successfully changed unimodally with ambient temperature (Figure 3A, GLM, χ²2=26.29, p<0.001, n=117).”

3) The authors mention that they included environmental covariates, elevation and daily minimum temperature, but did not indicate whether these had any influence on breeding success. Also, was there any attempt to account for year in the analysis?

Since elevation, mean air temperature, and daily minimum temperature were all highly correlated, elevation and daily minimum temperature were excluded from final analysis. We did not account for year effects previously, but to address this concern we have now added year as the random effect in the analysis, which makes the results even more clear (e.g. the optimal temperature of TPC with blowfly access becomes even lower (from 14.1 to 13.1℃, Figure 4A) compared with the ones without blowfly access (from 14.7 to 14.6℃, Figure 4B). We have modified the Materials and methods as follows:

“Other environmental factors (elevation, daily minimum air temperature) were fitted to test the generality of the results. […] We also included year as a random factor to account for sampling at different time points (See Source Code 1 for further details).”

4) The fact that the field group size experiments match the realized TPC in the field and the fundamental TPC in the lab exactly is curious, given how variable experiments (especially those in the field) can be. Perhaps the use of more significant figures and listing the uncertainty around these estimates would allay any concerns from readers.

We were also surprised by how closely these data match the optimal temperature between the realized TPC in the field and the fundamental TPC in the lab (the actual number is 15.64℃ for the lab and 15.59℃ for the filed results). We think this suggests that cooperation indeed can help burying beetles to breed at a temperature close to the physiological optimal temperature of the TPC.

5) Related to the above point: The code underlying the analyses should be included so readers can evaluate how the results were arrived at, also addressing concerns above regarding the model. (If it is not already planned to be included in the Dryad Repository, especially given the nice data break down by figure).

We have will certainly upload all of the final code with the completed manuscript.

6) How is competition between blowflies and beetles modeled in the theoretical section? There must be some equation used to represent how access to a carcass varies with temperature due to the activity of blowflies, but we couldn't see the equation or any reference to it.

We model the effects of interspecific competition by using different fundamental thermal performance curves (TPC) to represent the properties of competing species. Thus, rather than explicitly describing the detailed natural histories in the model, we instead described a scenario of “A low-temperature thermal specialist species resembles the burying beetle’s breeding thermal performance (Tsai, 2020), whereas a high temperature generalist species that has a similar life history to the blowfly (Figure 1A).” We also model the case of a high-temperature specialist competing with a low-temperature generalist in Figure 1—figure supplement 1.

7) Materials and methods: How many traps were placed within the altitudinal range, and how far apart were they?8) Origins of the lab strains of beetles used in the laboratory experiments: Please provide more details on how many lab strains were used in the experiments and the extent to which these lab strains originated from separate populations with distinct thermal conditions. Currently, the manuscript simply states that each lab strain derived from 2 males and 2 females caught in the field.9) Breeding performance in the common garden experiment: Please explain why males and females from different populations were paired up in this experiment, and whether this approach had any implications for the interpretation of the results. If lab strains originated from separate populations with distinct thermal conditions, and there is local adaptation to these thermal conditions, could this approach alter the offspring's TPC compared to that of natural populations? If so, could this have implications for the interpretation of your results, such as for comparisons of breeding performance under laboratory and natural conditions?10) Thermal regulation of locomotor performance: This experiment was conducted by beetles from different lab strains. Given that each lab strain derived from 2 males and 2 females caught in the field, there is likely to be some inbreeding in each lab strain. Could this have implications for the interpretation of these results, such as for comparisons of TPC for locomotory and breeding performance?

There was an error in how we described the source of our lab populations in our previous Materials and methods. We have two types of lab populations. In the primary type of lab population, we began trying to establish it in 2011 and it has become stable since 2014. The behavioral and physiological experiments of our lab are mainly conducted with this population, as we have described in Liu et al., 2020, and Chen et al., 2020. This is also the population we used for the lab experiments described in this paper.

This population is from mid-elevation on Mt. Hehuan. We have modified our description to as follows:

“Lab experiments were conducted using *N. nepalensis* individuals from the laboratory-reared population. […] Since establishment, we have supplemented the lab strain with new individuals from the same location every one or two years to avoid inbreeding.”

Our other type of lab population consists of five strains captured from five different locations in 2017 and 2018. This population was used specifically to test local adaptation as described in Tsai et al., 2020, but it was not used in this study. Our previous Materials and methods mistakenly described this lab population rather than our primary lab population, and we apologize for the confusion.

11) Breeding performance in the field: Our understanding is that different treatments were replicated (at least to some degree) in different years. The treatment 'with interspecific competition' was replicated in 2014 and 2015, whilst the treatment 'without interspecific competition' in 2014 and 2019? What implications, if any, could this have for the interpretation of your results?

We have modified this section to account for all of the data in this manuscript:

“In 2013 to 2016 (May-October), we investigated the natural pattern of *N. nepalensis* reproduction and its breeding success along an elevational gradient from 673m to 3422m on Mt. Hehuan in central Taiwan (24°11’ N, 121°17’ E) that encompasses broadleaf forests at lower elevations and mixed conifer-broadleaf forests at higher elevations.”

and also:

“Breeding experiments without blowflies were conducted in the same experimental sites from 2014 to 2017, and 2019 (May-October).”

We also modified the sentence in the Data Analysis part:

“We also included year as a random factor to account for sampling at different point.”

12) The Introduction (like the Abstract) doesn't emphasize, or rather weave together, the two important currents of niche research (TPC and cooperation, as mentioned above in the summary) that are addressed explicitly and simultaneously in this paper. As in the Abstract, it may be helpful to highlight cooperation (or competition/cooperation balance) earlier in the paper. Something akin to the gem of a paragraph that starts with "Integrating the idea of TPCs into the ecological niche concept helps bridge two rich, but largely independent, traditions of studying thermal adaptation." in the Discussion. A similarly strong argument earlier in the Introduction can help tie together the TPC/behavior currents.

Thank you for the suggestion. We have added the argument in the Introduction as follows:

“Furthermore, although a few studies have shown that intraspecific cooperation can also help social species expand their realized niche width (Sun et al., 2014; Lin et al., 2019), little is known about how intraspecific cooperation influences the realized TPCs of social organisms.”